# Soil Microbiome in Conditions of Oil Pollution of Subarctic Ecosystems

**DOI:** 10.3390/microorganisms12010080

**Published:** 2023-12-30

**Authors:** Elena N. Melekhina, Elena S. Belykh, Vladimir A. Kanev, Anastasia A. Taskaeva, Dmitry V. Tarabukin, Aurika N. Zinovyeva, Ilya O. Velegzhaninov, Elena E. Rasova, Olga A. Baturina, Marsel R. Kabilov, Maria Yu. Markarova

**Affiliations:** 1Institute of Biology, Komi Scientifc Center, Ural Branch of Russian Academy of Sciences (IB FRC Komi SC UB RAS), Kommunisticheskaya 28, 167982 Syktyvkar, Russia; belykh@ib.komisc.ru (E.S.B.); kanev@ib.komisc.ru (V.A.K.); taskaeva@ib.komisc.ru (A.A.T.); dim1822@mail.ru (D.V.T.); zinovyeva@ib.komisc.ru (A.N.Z.); vellio@yandex.ru (I.O.V.); rasova@ib.komisc.ru (E.E.R.); myriam@mail.ru (M.Y.M.); 2Institute of Chemical Biology and Fundamental Medicine, Siberian Branch of the Russian Academy of Sciences (ICBFM SB RAS), Lavrentieva 8, 630090 Novosibirsk, Russia; baturina@niboch.nsc.ru (O.A.B.); kabilov@niboch.nsc.ru (M.R.K.)

**Keywords:** oil pollution, bioremediation, metagenomic analysis, soils, stagnic cambisols, European North-East

## Abstract

The present study aimed to investigate the recovery of soil quality and the bacterial and fungal communities following various recultivation methods in areas contaminated with oil. Oil spills are known to have severe impacts on ecosystems; thus, the restoration of contaminated soils has become a significant challenge nowadays. The study was conducted in the forest–tundra zone of the European North-East, where 39 soil samples from five oil-contaminated sites and reference sites were subjected to metagenomic analyses. The contaminated sites were treated with different biopreparations, and the recovery of soil quality and microbial communities were analyzed. The analysis of bacteria and fungi communities was carried out using 16S rDNA and ITS metabarcoding. It was found that 68% of bacterial OTUs and 64% of fungal OTUs were unique to the reference plot and not registered in any of the recultivated plots. However, the species diversity of recultivated sites was similar, with 50–80% of bacterial OTUs and 44–60% of fungal OTUs being common to all sites. New data obtained through soil metabarcoding confirm our earlier conclusions about the effectiveness of using biopreparations with indigenous oil-oxidizing micro-organisms also with mineral fertilizers, and herbaceous plant seeds for soil remediation. It is possible that the characteristics of microbial communities will be informative in the bioindication of soils reclaimed after oil pollution.

## 1. Introduction

The impact of oil spills on ecosystems and the environment is significant, making the restoration of soil contaminated with petroleum products a contemporary challenge [1,2,3,4]. It is particularly pronounced in cold climates [5,6,7], where soil moisture and slow restoration processes contribute to the spread of oil products over long distances. Restoration efforts, including bioremediation, are employed to mitigate the effects of oil contamination on ecosystems. The selection of reclamation methods that are suitable for northern biocenoses is a practical priority. Research on the patterns of restoration in oil-contaminated ecosystems is crucial for diagnosing their condition and assessing the effectiveness of reclamation methods. In order to prevent further negative consequences, it is imperative to employ ecological remedies. Bioremediation methods are still supposed to be the most efficient, flexible, and environmentally conscious, despite requiring extended time to achieve decontamination benchmarks [3]. The inoculation of exogenous micro-organisms capable of utilizing oil products accelerates the adaptation of the indigenous population and its crude oil destruction ability [7], in particular, due to horizontal gene transfer [8]. It has demonstrated the efficacy of biopreparations containing indigenous micro-organisms [9,10], particularly when utilizing a complex of multiple strains due to the narrow specialization of certain bacteria and fungi for specific petroleum fractions [11]. Additionally, the simultaneous application of biopreparations with high plant seeds has been found to be an effective strategy [2,6,9,12,13]. The use of mineral fertilizers in conjunction with biopreparations is another crucial determinant of the speed and efficiency of bioremediation, according to prior studies [9,14,15]. Previous research has focused on conducting comprehensive dynamic studies of phytocenoses, zoocenoses, and microbiocenoses in oil-contaminated soils in the Far North [10,16]. A long-term monitoring study was carried out to analyze the effects of various bioremediation methods on the restoration of biotic and abiotic components in contaminated soil [10,16]. Additionally, research has begun to evaluate the microbial diversity of oil-contaminated soils using metagenomic analysis [17]. Currently, active research is being conducted on soil microbial diversity based on metagenomic analysis [18]. Metagenomic monitoring is used to assess the disturbed soils’ condition [19]. The present study is unique in that it represents the first time the genetic diversity of micro-organisms has been examined in relation to the assessment of bioremediation methods for oil-contaminated soils in the Far North. The objective of this study was to investigate soil quality and the recovery of bacterial and fungal communities following the application of different recultivation procedures to the oil-contaminated area.

## 2. Materials and Methods

### 2.1. Study Area

The sites studied are located at the Verkhnevozeyskoe oil field (Usinsky district, Komi Republic: 66°37′40″ N, 57°07′56″ E) (Figure 1), where oil spills occurred several times between 1989 and 1996 [9]. The study area is located in the forest–tundra zone and characterized by harsh climate conditions: low temperatures, strong winds, and a short growing season [20]; soils—stagnic cambisols [21].

The present study was conducted at experimental recultivation site number 20, which was established in the territory of severe oil pollution to investigate the efficacy of different remediation techniques using various biological products. The bioremediation experiment described in detail in [9,22] was initiated in 2002, 13 years after the occurrence of the oil spill. The study site was subjected to preparatory measures prior to recultivation which included removal of oil from the surface; drainage of excess water; and plowing up to a depth of 25–30 cm. The territory was divided into nine sites where several biorecultivation techniques were tried out. In 2019, soil samples were taken from six of these sites for metagenomic and chemical analysis (the original numbering of the sites was retained). Site 2 did not undergo any recultivation measures, while sites 4, 6, 7, and 9 were treated with biopreparations, herb seeds, and fertilization (Table 1). Site R is the reference site located in similar edaphotopic conditions with other studying sites, but it remained unaffected by oil contamination and other anthropogenic influences.

The biological product “Universal” containing yeasts *Rhodotorula glutinis* and bacteria *Rhodococcus egvi*, *Rhodococcus erythropolis*, and *Pseudomonas fluorescens* [23] was applied at site 4 and site 6. Furthermore, lignin sorbents consisting of dried and crushed compost with embedded oil-destructing micro-organisms isolated from the oil-contaminated soils of the study area, and biologically active fertilizer-seeding granules (BAG) on the compost basis with fertilizers and seeds of perennial herbs were added to the soil at site 6 [9]. At site 9, the biological product “Roder” was added to the soil. “Roder” comprises bacteria including *Rhodococcus ruber* and *Rhodococcus erythropolis* [24]. Mineral fertilizers in the form of ANP (ammonium nitrate phosphate) were utilized at a rate of 350 kg/ha across all sites, with the exception of site 2. The introduction of the fertilizer occurred in 2002, with no subsequent additional feeding. Moreover, at site 7, dolomitic meal was employed in conjunction with the mineral fertilizers [9].

**Plant communities.** Site 2 represents sedge–mixed herbs community (Figure 2). The tree layer consists of Siberian spruce (*Picea obovata*) reaching heights of up to 0.5 m, and downy birch (*Betula pubescens*) up to 1 m. The bush layer includes downy willow (*Salix lapponum*) and willow filicifolia (*Salix phylicifolia*) up to 1 m. The grass–bush layer has a total projective cover (TPC) of 60–70%. Bilberry (*Vaccinium myrtillus*), bog bilberry (*Vaccinium uliginosum*), crowberry (*Empetrum hermaphroditum*), marsh Labrador tea (*Ledum palustre*), and dwarf birch (*Betula nana*) were registered among shrubs. Creeping sedge (*Carex chordorrhiza*) with projective cover (PC) 30–40%, common cottongrass (*Eriophorum polystachion*) with PC 7–9%, and tussock cottongrass (*Eriophorum vaginatum*) with PC 10–15% are dominant species in the studied area. Additionally, narrow-leaved willow-herb (*Chamaenerion angustifolium*), common club moss (*Lycopodium clavatum*), and northern firmoss (*Huperzia selago*) are prevalent herbaceous species in the area. The moss cover with TPC of 20–25% is primarily composed of *Polytrichum commune* and *Calliergon giganteum*.

Site 4 was identified as a mixed herbs–hairgrass community. The tree layer was presented with downy birch (*Betula pubescens*) 1–1.5 m (approximately 30 individuals) and willow filicifolia (*Salix phylicifolia*) 1 m (about 10 individuals). The grass layer with TPC of 70–80% consisted of tufted hairgrass (*Deschampsia cespitosa*) (PC 40–50%), tussock cottongrass (*Eriophorum vaginatum*) (PC 10–15%), common cottongrass (*Eriophorum polystachion*) (PC 3–5%), black bent (*Agrostis gigantea*), narrow-leaved willow-herb (*Chamaenerion angustifolium*), reed canary grass (*Phalaroides arundinacea*), gray sedge (*Carex cinerea*), and purpur reed grass (*Calamagrostis purpurea*). The moss cover with TPC of 20–30% was dominated by green mosses *Polytrichum* sp.

Site 6 is characterized as mixed herbs–hairgrass community. Grass layer with a TPC of 70–80% consists of tufted hairgrass (*Deschampsia cespitosa*) with a PC of 50–60%, purpur reed grass (*Calamagrostis purpurea*), black bent (*Agrostis gigantea*), narrow-leaved willow-herb (*Chamaenerion angustifolium*), and a moss cover with a TPC of 10–15% including green mosses such as *Polytrichum commune* and *Calliergon giganteum*.

Site 7 was identified as mixed herbs–hairgrass community. The tree layer was presented with downy birch (*Betula pubescens*), reaching heights of 1.5–2 m (about 30 individuals); the bush layer is comprised of willow filicifolia (*Salix phylicifolia*) 1.5 m (about 30 individuals). Grass–bush layer accounts for 70–80% of TPC and includes species such as crowberry (*Empetrum hermaphroditum*) and tufted hairgrass (*Deschampsia cespitosa*) with PC of 30–35%, reed canary grass (*Phalaroides arundinacea*) with PC of 10–15%, purpur reed grass (*Calamagrostis purpurea*) with PC of 10–12%, common cottongrass (*Eriophorum polystachion*), tussock cottongrass (*Eriophorum vaginatum*), narrow-leaved willow-herb (*Chamaenerion angustifolium*), and black bent (*Agrostis gigantea*)—among herbs. The moss cover, representing 20–30% of TPC, includes various green moss species.

Site 9 is characterized as a canary grass–mixed herbs community, with a tree layer consisting of downy birch (*Betula pubescens*) 1–1.5 m (about 30 individuals), Siberian larch (*Larix sibirica*) 2 m (1 individual), common pine (*Pinus sylvestris*) 2 m (2 individual), and Siberian spruce (*Picea obovata*) 1.5 m (1 individual). The grass layer with TPC of 50–60% was dominated by reed canary grass (*Phalaroides arundinacea*) with PC 20–30%, black bent (*Agrostis gigantea*) with PC 5–10%, beak sedge (*Carex rostrata*) with PC 5–10%, tufted hairgrass (*Deschampsia cespitosa*), narrow-leaved willow-herb (*Chamaenerion angustifolium*), hedge bedstraw (*Galium mollugo*), marsh willowherb (*Epilobium palustre*), purpur reed grass (*Calamagrostis purpurea*), and common cottongrass (*Eriophorum polystachion*). The moss cover with TPC of 20–30% included several liverworts and *Pleurozium schreberi*.

Thus, all contaminated sites that have been studied exhibit a well-defined herbaceous community composed of representatives, with a grass cover of more than 50%. Grass species, including tufted hairgrass (*Deschampsia cespitosa*), reed canary grass (*Phalaroides arundinacea*), and black bent (*Agrostis gigantea*) predominated on sites 4–7, while sedges were dominant on site 9. Woody vegetation on most contaminated sites consisted of single young trees of downy birch (*Betula pubescens*), as well as some conifers such as *Pinus sylvestris*, *Picea obovata*, and *Larix sibirica*, along with some willow species (*Salix* sp.). The projective cover of mosses ranged from 15 to 30%.

The reference site presented a community of mixed dwarf birch–shrub sphagnum open forest. The tree layer included Siberian spruce (*Picea obovata*) 5 m, downy birch (*Betula pubescens*) 4 m, bushes–dwarf birch (*Betula nana*), willow filicifolia (*Salix phylicifolia*), and gray willow (*Salix glauca*) 1 m, with PC 70–80%. Grass–bush layer has TPC of 60–70%. Shrubs included crowberry (*Empetrum hermaphroditum*) with PC of 20%, bog bilberry (*Vaccinium uliginosum*) with PC of 20%, bilberry (*Vaccinium myrtillus*) with PC 10%, marsh Labrador tea (*Ledum palustre*) with PC 10%, grass layer–globular-spike sedge (*Carex globularis*) with PC 15%, and field horsetail (*Equisetum arvense*). Moss cover with TPC 90% had *Sphagnum* sp. with PC 60%, *Polytrichum commune* with PC 10–5%, and mixed green mosses.

At present, the experimental site is inhabited by several herbaceous communities, namely, forb–pike grass, canary grass–forb, and sedge–forb. On most sites (except site 2) the species composition of these communities is generally comparable, characterized by the dominance of *Phalaroides arundinacea*, *Deschampsia cespitosa*, *Agrostis gigantea*, and *Calamagrostis purpurea*, along with some water-loving representatives of the sedge family, such as polystachion cotton grass (*Eriophorum polystachion*), vaginal cotton grass (*Eriophorum vaginatum*), and bottle sedge (*Carex rostrata*), which have colonized the territory with the aid of local flora. At the beginning of the experiment, sedges were not initially observed. In three out of the five experimental sites, the dominant plant species was *Deschampsia cespitosa*. Among other herbaceous plants, the most commonly noted are fireweed (*Chamaenerion angustifolium*), soft bedstraw (*Galium mollugo*), and marsh fireweed (*Epilobium palustre*). Site 2 resulted in the formation of a community, that closely resembled natural background communities found in swamps, peat bogs, and swampy forests. This community included shrubs such as *Vaccinium myrtillus*, *Vaccinium uliginosum*, *Empetrum hermaphroditum*, *Ledum palustre*, and *Betula nana*, as well as herbaceous plants—polystachion cotton grass (*Eriophorum polystachion*), vaginal cotton grass (*Eriophorum vaginatum*), and whip sedge (*Carex chordorrhiza*). In those sites with high herbaceous plant participation, as indicated by a TPC more than 70–80%, and where *Deschampsia cespitosa* dominates (sites 6 and 9), the woody plants have low participation. This is due to the high density of the herbaceous layer, which hinders the penetration of woody plants. Conversely, in sites with sparse grass cover or low participation of *Deschampsia cespitosa*, the number of woody plants increases (site 7). In such areas, the woody plant participation can be substantial, with the trial area having over 100 individuals. At the outset of the experiments, all sites lacked moss cover, which began to appear after four years. Currently, the projective cover has increased to 30% in most sites.

### 2.2. Soil Sample Collection

Soil samples were collected from all four corners and the center of the 0.2 × 0.2 m site, combined and thoroughly mixed. For analysis of soil fungal and bacterial communities, a total of 39 samples were collected, representing a mixture obtained from a 15 cm-deep soil column. The samples were flash-frozen in liquid nitrogen and stored at −80 °C until analysis.

### 2.3. Soil Chemical Analyses

Soil pH was determined using a 1:2.5 (*w*:*v*) ratio of soil to deionized water [25]. The organic C and N total contents were determined using an EA-1100 analyzer (CHNS-O, CE Instruments, Pisa, Italy); FR.1.31.2016.23502), and total petroleum hydrocarbons (TPH) using gravimetry. The heavy metals chemical analysis in the soil was performed by inductively coupled plasma atomic emission spectroscopy. The chemical analysis of the soil was performed in “Ecoanalit” laboratory (Accreditation number RU.0001.511257) of the Institute of Biology (Komi Science Center, Urals Branch of the Russian Academy of Sciences).

### 2.4. Soil Metabarcoding Analysis

Metabarcoding analysis was conducted on 39 soil samples collected from six sites described above. Total DNA was extracted using the DNeasy PowerSoil Kit (Qiagen, Germantown, MD, USA) as per manufacturer’s instructions. The bead beating was performed using TissueLyser II (Qiagen, USA) for 10 min at 30 Hz. The quality of the DNA was assessed using agarose gel electrophoresis.

The 16S rRNA gene and ITS2 regions were amplified with the primer pairs V3/V4 and ITS3_KYO2/ITS4, respectively, combined with Illumina adapter sequences [26]. PCR amplification was performed as described earlier [27]. A total of 200 ng PCR product from each sample was pooled together and purified through MinElute Gel Extraction Kit (Qiagen, Hilden, Germany). The obtained amplicon libraries were sequenced with 2 × 300 bp paired-ends reagents on MiSeq (Illumina, San Diego, CA, USA) in the SB RAS Genomics Core Facility (ICBFM SB RAS, Novosibirsk, Russia).

Raw sequences were analyzed with UPARSE pipeline [28] using Usearch v11.0.667. The UPARSE pipeline included paired-reads merging; read quality filtering; length trimming; identical-reads merging (dereplication); discarding singleton reads; removing chimeras; and OTU clustering using the UPARSE-OTU algorithm [29]. The OTU sequences were assigned a taxonomy using the SINTAX [30] with 16S RDP training set v16 [31] and fungi ITS UNITE v.8.2 [32] as references. As a result of the analysis performed, there were 1,385,659 reads of the 16S RNA gene fragment and 947,173 annotated reads of the ITS.

Alpha diversity metrics were calculated in Usearch. Rarefaction and extrapolated curves were generated using the “iNEXT” package [33]. The Mann–Whitney test was performed using the Python scientific computing library, SciPy (v.1.5.1).

### 2.5. Statistical Analyses

Statistical processing of the data was carried out using Microsoft Office Excel (Microsoft Office, Redmond, WA, USA) and the R programming language [34]. The table shows mean values and standard deviations. The values of the Chao-1 and Shannon α-biodiversity indices calculated for each individual sample were averaged to obtain the value characterizing the site. All values of the fraction reads annotated for one or another OTU were also averaged among the samples of one site. Differences in α-biodiversity indices of experimental sites and differential abundances were assessed using the Mann–Whitney test. All multiple comparisons were performed with false discovery rate (FDR) correction. Those OTUs that accounted for more than 1% of all 16S rDNA or ITS reads were considered dominant. A nonmetric multidimensional scaling (NMDS) analysis was performed with the R programming language [34,35].

## 3. Results

### 3.1. Chemical Properties of the Soil

The chemical analysis results (Table 2) indicate that the pH values of the soil on the sites studied are higher than typical for the forest–tundra soils of the region [21]. The acidity of sites 2 and 9 are even significantly higher than that of the reference site. Despite the addition of mineral fertilizers to the soils of all sites, except site 2 during the reclamation of disturbed area, and the development of plant meadow communities [22], a deficiency of minerals, including N and Zn, is still registered. There were no excesses of the regional background concentrations of toxic Pb and Cd [21] detected.

In general, the concentration of petroleum hydrocarbons in the surface layer of the reference site is 0.9 mg/g of soil, which is slightly lower than the permissible level [36], and, formally, this area is not considered polluted. However, due to its proximity to oil spills and the lateral transport of hydrocarbons, the TPH level here is significantly higher than the background one [37] determined as 11–32 mg/kg.

The level of soil contamination with oil products at experimental sites at the time of sample collection can be still assessed as extremely high [37]. However, during the observation period, the level of petroleum products in the surface layer decreased, on average, by 3–7 times [22]. The maximum reduction is manifested for site 6 (up to 33 times) where a comprehensive approach to remediation was applied: plant seeds and biopreparation were added in a special granulated form.

In addition to the total concentrations of TPH in the soil, the levels of several highly toxic [38] polycyclic aromatic hydrocarbons were measured. Among them were Benz(a)anthracene, Chrysene, and Phenanthrene (Figure 3). Overall, significant differences between sites were not observed, but exact concentrations in some individual points were extremely high, while, in others, they were below the detection limit.

### 3.2. Bacteria Diversity

The reference site indicated the highest average number (1402) of distinct bacteria OTUs out of 4975 OTUs registered in the area studied. Conversely, the sites contaminated by oil showed lower OTU numbers, ranging from 556 at site 4 to 1153 at site 9. Analyses of diversity indices (Figure 4) confirmed a significant decrease in bacteria diversity at sites 4 and 6. The Chao index values were reduced at sites 4, 6, and 7, while Shannon’s diversity index values were decreased at sites 4 and 6.

Representatives of Acidobacteria and Proteobacteria comprise approximately 50% of the bacterial community in reference sites (Figure 5) and over 75% in oil-contaminated sites. Notably, Acidobacteria is the dominant phylum, representing about 50% of the community on sites 4 and 6, while the abundances of both phyla are similar on the other sites. Furthermore, it is worth noting that, when considering the number of different OTUs, Proteobacteria OTUs are identified 2–2.5 times more frequently than Acidobacteria OTUs.

Representatives of other phyla, including Actinobacteria, Bacteroidetes, candidatus Saccharibacteria, and especially Chloroflexi, were found to be significantly more abundant on the reference site as compared to the contaminated ones. Gemmatimonadetes exhibited equal abundances on the reference site and site 9, but with a much lower abundance on other contaminated areas. On the contrary, Verrucomicrobia were higher in the contaminated area, with abundances reaching up to 2–2.5 times those of sites 4, 6, and 7. Cyanobacteria abundance was only found to be higher than 2% on site 6 and insignificant on others. The abundances of candidate division WPS-1 OTUs were similar across all the sites.

A total of 31 phyla were identified in the area studied. The phyla Armatimonadetes, BRC1, wps-2, Chlamydiae, Firmicutes, Ignavibacteriae, Latescibacteria, Parcubacteria, Phycisphaerae, and Planctomycetes were found to have relatively low abundances. In contrast, the remaining phyla were each represented by only 1–5 OTUs.

The bacterial communities’ diversity in areas contaminated with crude oil remains notably lower than that of the reference site. An analysis of the top 20 OTUs with maximal abundances, as depicted in Figure 6, reveals that 13 out of these 20 OTUs belong to Acidobacteria. The remaining five OTUs are associated with Proteobacteria, while the last is assigned to the unknown Subdivision 3 genera incertae sedis, which is abundant in a majority of the samples analyzed. Moreover, the abundances of these top 20 OTUs vary significantly across the contaminated sites. The absence of six to ten representatives of the top 20 OTUs at the reference site is noteworthy, indicating a significant divergence in community composition. Furthermore, a majority of the remaining OTUs display low abundances, with only two OTUs persisting as dominant across all sites under study. Moreover, site 4 exhibits the complete absence of one to seven OTUs from the top 20 list, while simultaneously demonstrating maximal abundances among the soil samples. The distribution of the top 20 OTUs across the other sites is relatively even, with their abundances falling within the average range. However, exceptions are observed in OTU_6 (unc_Mycobacterium) and OTU_7 (unknown Acidobacteria), which display a low abundance at sites 2 and 9.

The prevalence of several OTUs on the contaminated sites with crude oil and its decomposition products is a clear indication of their association with the presence of these contaminants. The distinct position of the reference site is supported with the analysis of OTUs that are unique to each site (Figure 7). The reference site registered the highest number of OTUs (3294), with more than half of them (68%) being unique to that site and not found on the oil-contaminated sites. Despite the different recultivation methods employed, all contaminated sites exhibit a high degree of similarity, sharing between 50 to 80% of their OTUs. Unique to the oil-contaminated sites community are 1278 OTU that are not found at the reference site. About half (459) of OTUs present exclusively on the oil-contaminated sites belong to Proteobacteria, followed by Bacteroidetes (106) and Acidobacteria (77).

The results of ordination analysis (Figure 8A), based on Bray-Curtis distance metric, provide additional evidence for the distinctive positioning of the reference site. The samples taken from oil-contaminated sites exhibit spatial proximity to one another. The application of constrained statistical methods (Figure 8B) reveals a substantial correlation between soil concentrations of nitrogen (*p* = 0.005), heavy metals (Cu, Zn, Pb and Cd, *p* = 0.01), crude oil (*p* = 0.02) and soil pH (*p* = 0.01). The distribution of contaminated site points on the ordination plot forms a tightly clustered group, in stark contrast to the reference site points. This discrepancy may be attributed to the noted disparities in nitrogen content and crude oil level between the soils of the reference site and contaminated ones. The reference samples (Figure 8C) indicate a distinct disparity in their eigenvalues, with only pH (*p* = 0.005) and soil NO_3_ content (*p* = 0.05) demonstrating statistical significance. Furthermore, the contaminated sites display an uneven distribution of bacterial diversity, as the points of site 2 that underwent solely mechanical cleaning of crude oil and site 4 which is typified by its conspicuously low diversity, form divergent clusters. Thus, diversity of Bacteria in oil-contaminated sites is shown to be influenced substantively by soil fertilization and biopreparation methods as opposed to current levels contamination.

### 3.3. Fungi Diversity

The results of the analysis revealed that fungi abundance on site 4 was notably lower in comparison to the other sites, as evidenced by the calculations of the Shannon index. However, the observed decrease in the Chao index was not significant (Figure 9). A comprehensive survey of the studied area confirmed the presence of representatives from all 14 Fungi phyla. Remarkably, the majority of fungal representatives, ranging from 60 to 80 on average, across all sites belonged to Ascomycota (Figure 10). Following closely behind was Basidiomycota, with the exception of site 7 where an approximately equal proportion of Ascomycota and Basidiomycota OTUs were observed. The other phyla accounted for less than one % of the overall fungal diversity. Representatives of Rozellomycota, Glomeromycota, Mortierellomycota, Chytridiomycota, Zoopagomycota, Zygomycota, and Mucoromycota phyla were registered across all sites. On the other hand, Aphelidiomycota, Olpidiomycota, Basidiobolomycota and Blastocladiomycota were exclusively recorded in the reference samples. The proportion of unidentified fungi was exceptionally.

If based solely on diversity index calculations and the most abundant phyla distribution, an incomplete restoration of Fungi communities in oil-contaminated area studied even 30 years of the oil spill, could be inferred. Also, the detailed examination of the soil Fungi structure indicated its heterogeneity. Among the top 20 ITS with the highest abundance, it was unsuprising to detect 14 Ascomycota and six Basidiomycota representatives (Figure 11). Nevertheless, the distribution of OTUs comprising the top 20 amongst the soil samples depicts an extraordinary degree of heterogeneity. The majority of samples contained only two OTUs while the remaining OTUs were abundant in a few samples and scarce or non-existent in others. Among the top 20 OTUs, four were found to be prevalent in samples from the reference site (Figure 11). It is particular significance that OTU6, representing the Rhodotorula genus, was registered in the top 20 OUTs across all oil-contaminated sites that underwent biological recultivation, while this OTU was largely absent in samples from site 2 and the reference area. Furthermore, sites 4, 6 and 7 displayed a closer relationship to each other than with site 9, within the context of biorecultivated sites.

Venn diagrams (Figure 12) provide whole support for the isolated position of the reference site. A significant portion, specifically 64% of OTUs, which amounts to 563 out of 875, are unique to this site. Similar to bacterial communities, fungal communities in oil-contaminated sites exhibit a lower number of OTUs and a greater similar with each other compared to the reference site community. The reference site boasts the highest number of OTUs at 875, followed by site 9 with 744 OTUs, while the remaining contaminated sites had comparable OTU numbers. Nonetheless, over half of the reference site OTUs (64%) are unique, with no representation in the oil-contaminated sites. Similar to the situation with bacterial communities, contaminated sites, regardless of the recultivation method employed, indicate a commonality of 44 to 60% of OTUs, that is twice the level of similarity between the reference area and each of the contaminated sites.

According to the Bray–Curtis distance ordination (Figure 13), it is observed that all the sites subjected to biological recultivation in the presence of oil contamination exhibit a tendency to aggregate together within the sample space. This clustering phenomenon is further substantiated by the isolated positioning of the reference site and site 2, wherein the latter has encountered mechanical reclamation solely targeting the upper soil layer. Furthermore, a concomitant investigation into the bacterial communities revealed the emergence of notable associations with the soil chemical characteristics, as substantiated when utilizing the constrained method (Figure 13B). The sample ordination is determined based on various soil properties, including NO_3_ content (*p* = 0.005), heavy metals (Cu, Zn, Pb and Cd, *p* = 0.01), and crude oil (*p* = 0.015). It is shown that points belonging to contaminated sites form a densely clustered group, while points corresponding to the reference site are located separately from the other samples, indicating considerable spatial separation. Upon excluding the reference samples from the analysis (Figure 8C), it becomes evident that soil factors such as NO_3_ content (*p* = 0.005), pH (*p* = 0.01), and PAH level (*p* = 0.01) play significant roles affecting the ordination patterns of the samples. Additionally, similar to the analysis of bacteria diversity, points representing site 2, which underwent the mechanical cleaning of crude oil, form a separate group.

## 4. Discussion

The article presents the results of an investigation of the efficiency of various biological reclamation methods for restoring crude-oil-contaminated areas in Far North. Over the course of more that 20 years, a significant decrease in soil contamination with crude oil was registered. The most noticeable effect was achieved through the use of an integrated approach, which involved the application of biopreparations in granular form. The maximum decrease in the level of petroleum products (up to 33 times lower than the initial levels) (Table 2) was recorded in area 6, where the biological product “Universal” was used based on micro-organism oil degraders isolated from the soils of the study area. Previously, we concluded that the use of the biopreparations “Universal” and “Roder” leads to a more rapid soil remediation from oil contaminants during the early stages of bioremediation (4 and 7 years post-soil treatment), compared to other soil remediation techniques [10]. In areas using these biological products, an increase in soil enzymatic activity, an increase in the number of oil-oxidizing micro-organisms, and an increase in the abundance of soil microarthropods were observed [10].

Nevertheless, even after a considerable duration since the occurrence of the oil spill, there is no conclusive evidence of community recovery. This could possibly be attributed to the extremely high levels of oil products in the soil at the beginning of the experiment. Several investigations [39,40,41] highlight that the restoration of oil-contaminated soil using biopreparations depends on the initial concentration of hydrocarbons. The efficacy of the biopreparation is most pronounced in cases of low initial pollution levels, owing to the inherent toxicity of oil products, especially PAH, as well as the chemically hydrophobic nature of crude oil [38,39].

This leads to a strong binding of oil to soil particles and a decrease in the effectiveness of its biodegradation due to the disruption of the soil water–air regime. In addition, a single application of mineral fertilizers during reclamation is insufficient. It is known that for the activation of oil oxidation by natural micro-organisms, a constant presence of nitrogen in the soil is required. The simultaneous application of mineral fertilizers and bacterial preparations leads to a sharp increase in oil biodegradation processes [42] due to an increase in the number of oil-oxidizing micro-organisms and activation of the entire microbial pool [42,43].

Our research indicates that the composition of microbial communities analyzed at the dominant phyla level exhibited similarities to those observed in other sites contaminated with oil [17]. Significant differences in the microbial community structure were observed between the experimental sites and the reference site (Figure 8 and Figure 13). In the reference site, certain microbial groups were found to be more abundant compared to the experimental site (Figure 5 and Figure 10). According to modern ideas of the remediation of oil-contaminated communities, the role of soil bacteria and fungi in the decomposition of hydrocarbons is typically less than 0.1% in areas that are not contaminated by petroleum products, but can increase to 10% in the first two years after contamination [3]. At the same time, the abundance and diversity of indigenous micro-organisms decrease [6] as a result of oil toxicity and changes in soil structure and properties. The number of oil-degrading species decreases with the time and the indigenous bacterial community is restored [6,8,44]. The horizontal transfer of genes has a great impact in increasing the diversity and ability of bacterial communities to degrade different hydrocarbons in oil-contaminated soils [8]. An increase in the carbon fraction within the C:N ratio enhances the rate of the horizontal transfer [8].

A decrease in the diversity of both bacterial and fungal communities has been observed in contaminated areas, but significant differences were only found in the bacterial communities of sites 4 and 6 compared to the reference site (Figure 4 and Figure 9). This may likely be due to the indirect effects of biological products on the natural soil microbiome, which may result in changes in biodiversity. Fungi species *Rhodotorula glutinis* inoculated to sites 4 and 6 as a biopreparation was not currently registered. Several representatives of bacteria and fungi known as petroleum destructors [1,3,44] are presented at the contaminated sites with low abundance, insignificant compared to the reference area. Eight OTUs of the *Rhodotorula* genus have been identified, with some known to be hydrocarbon-degrading micro-organisms (among them, indigenous *R. glacialis*, *R. ingeniosa*, *R. cresolica,* and *R. muscilagenosa*) [45]. Additionally, species of Proteobacteria, such as *Syntrophorhabdus aromaticivorans* [46], and Acidobacteria, such as *Holophaga foetida* [47], have been determined on oil-contaminated sites as polycyclic aromatic hydrocarbon (PAH)-degrading bacteria.

The differences in bacteria communities between the reference and contaminated sites were determined mainly by an increase in the proportion of Acidobacteria (Figure 5 and Figure 6), one of the most typical and widely distributed bacteria phylum in different soils [48,49,50,51]. These oligotrophic bacteria are adapted to low nitrogen environments and their diversity and abundance decrease [48] with an increase of this element in soil. Despite the frequent correlation of *Acidobacteria* abundance with specific pollutants, there is a lack of data supporting their direct involvement in pollutant degradation activities [52].

The differences in fungi communities most probably resulted from the mycorrhizal species diversity and abundance (Figure 10 and Figure 11). For instance, on the reference site, 13 from 19 representatives of the Sebacinales family have been identified, indicating their ability to form mycorrhizal associations with both deciduous and coniferous trees, as well as Marchantiophyta [53]. In contrast, contaminated sites show a much lower presence of this family, typically ranging from 1 to 5 OTUs. The abundance of OTU76 on the reference site is 25 times higher than that found on contaminated sites. Similar patterns were observed for representatives of the Glomeromycetes order, able to form mycorrhizae with a variety of vascular plants and Bryophyta thallus. The reference site showcases the presence of eight Diversiporales OTUs and 32 Glomerales OTUs. Some of the Glomerales were also found on sites 4–9; their abundance and diversity were significantly lower. Notably, site 2 does not harbor any Glomeromycetes.

It is noted that, in 2019, 20 years after soil remediation, the structure of microbial communities in site 2 differed from that of other recultivation sites (Figure 7, Figure 8, Figure 11 and Figure 12). On the one hand, this may be due to the method of reclamation, namely, only the technical removal of oil (Table 1), and, on the other hand, to the moisture regime, which is characterized by the appearance of plants that prefer waterlogged habitats.

In site 2, a sedge–mixed herbs community was developed; the dominant plants included the next moisture-loving species creeping sedge (*Carex chordorrhiza*), common cottongrass (*Eriophorum polystachion*), and tussock cottongrass (*Eriophorum vaginatum*). The plants that prefer conditions of high soil moisture, such as bog bilberry (*Vaccinium uliginosum*), crowberry (*Empetrum hermaphroditum*), and marsh Labrador tea (*Ledum palustre*), were also registered.

In addition, the results obtained in Figure 8 and Figure 12 are confirmed by literature data, indicating that some fungal populations can coexist, occupying different moisture niches, and the high plasticity of fungal communities allows them to be classified as more sensitive indicators of soil moisture than bacteria [54].

Thus, new data obtained through soil metabarcoding confirm our earlier conclusions about the effectiveness of using biopreparations with indigenous oil-oxidizing micro-organisms for soil remediation. It is possible that the characteristics of microbial communities will be informative in the bioindication of soils reclaimed after oil pollution.

## 5. Conclusions

The integrated approach usage, including the simultaneous application of mineral fertilizers, herbaceous plant seeds, and biological products in the form of granules, has proven to be the most effective strategy for the remediation of oil-contaminated soils in the Far North. The granulated form provided seedlings and micro-organisms with the required nutrition at the initial stages of growth.

The diversity of soil fungi on all the contaminated sites and bacteria on sites 2 and 9, at present, do not differ from the reference site and are similar in phyla composition for all sites. However, a detailed examination still shows a close similarity among the contaminated sites and a difference with the reference one that could be explained with the lack of nutrients. Moreover, the uncompleted succession of high plant communities on the contaminated sites should impact the soil microbiome structure.

The new data obtained on the microbiota of oil-contaminated soils can be applied in the biodiagnostics of reclaimed soils. They can be used as an addition to the system of criteria proposed earlier [10,16] to assess the effectiveness of remediation methods in the Far North.

## Figures and Tables

**Figure 1 microorganisms-12-00080-f001:**
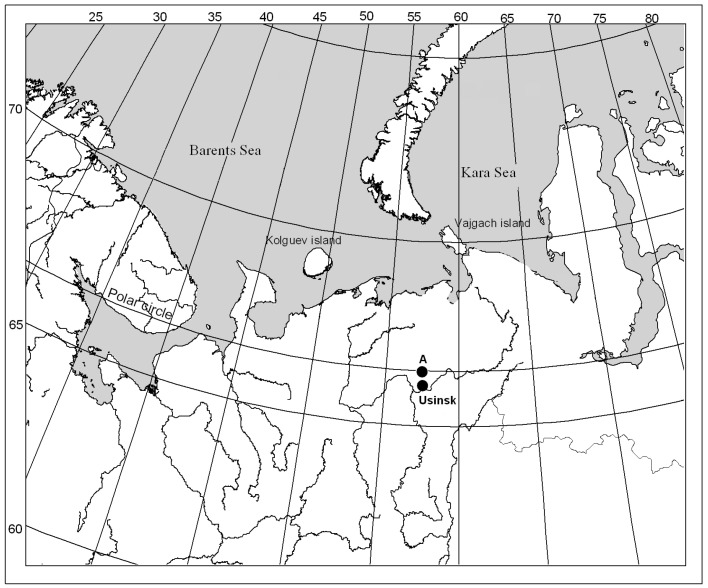
Usinsk city location and the experimental site (A).

**Figure 2 microorganisms-12-00080-f002:**
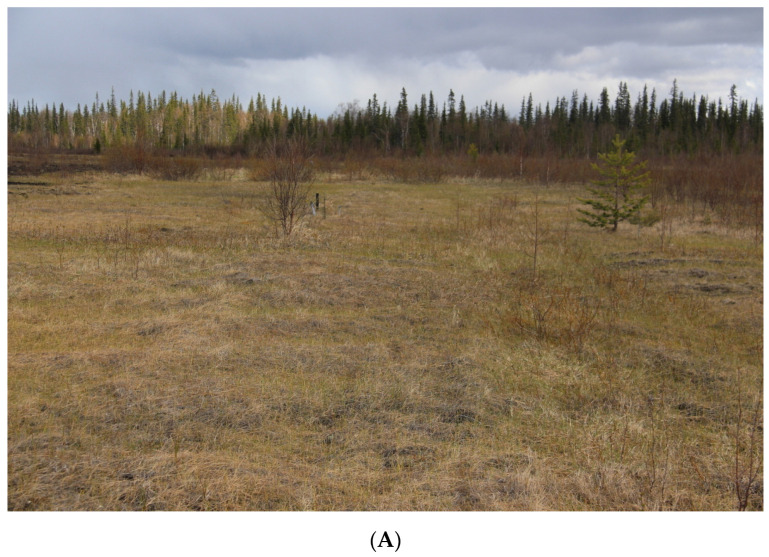
Experimental sites during the observation period (June 2019). (**A**) Site 2. Sedge-mixed herbs community. (The site exhibits natural regeneration, transitioning from a state of complete vegetation absence to a community dominated by sedges (*Carex* sp.) and cotton grass (*Eriophorum* sp.).); (**B**) Site 4. Mixed herbs–hairgrass community. (Predominance of turfed hairgrass (*Deschampsia cespitosa*), accompanied by the presence of other grass species in the herbaceous layer.); (**C**) Site 6. Mixed herbs–hairgrass community. (Turf pike complete dominance (*Deschampsia cespitosa*) over other species; the other species participation is not significant.); (**D**) Site 7. Mixed herbs–hairgrass community. (Soddy pike dominance (*Deschampsia cespitosa*) together with other grasses—reed grass (*Phalaroides arundinacea*) and purple reed grass (*Calamagrostis purpurea*); in addition, numerous young trees are represented.); (**E**) Site 9. Canary grass–mixed herbs community. (*Phalaroides arundinacea* dominance with the other grass species participation, sedges and various grasses with isolated young coniferous trees).

**Figure 3 microorganisms-12-00080-f003:**
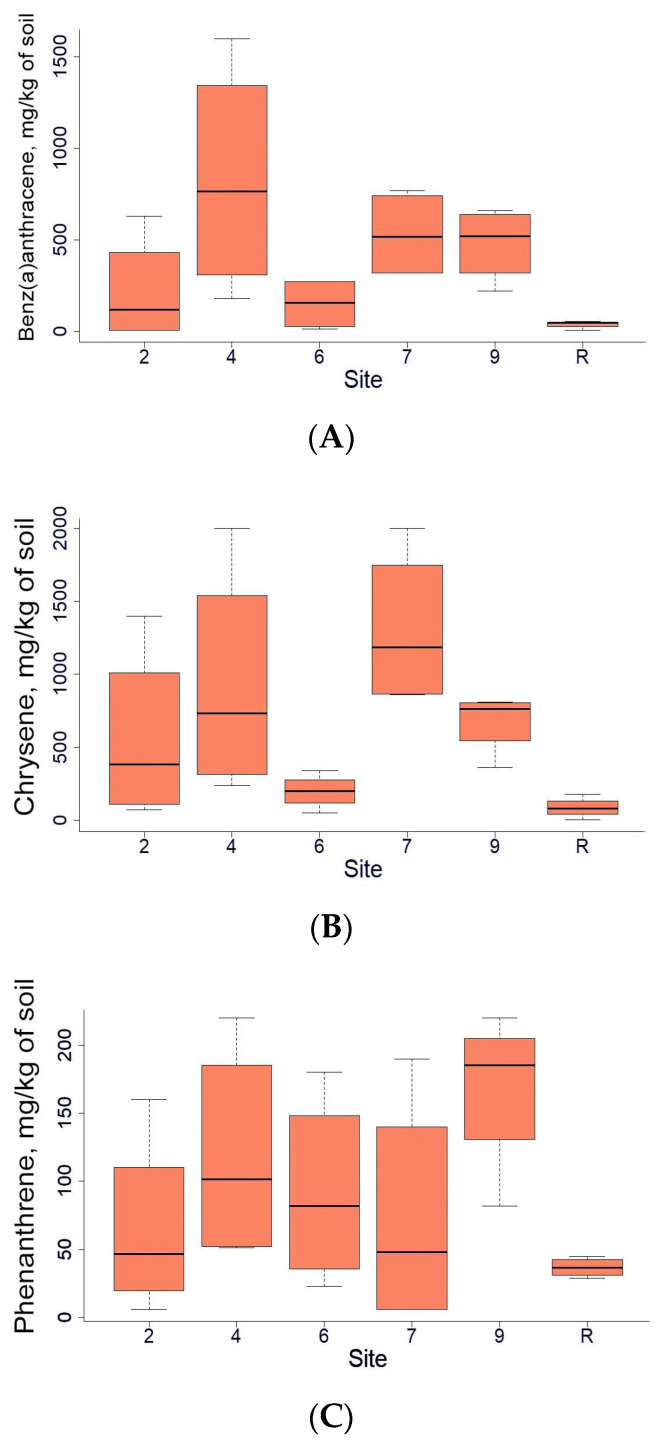
Benz(a)anthracene (**A**), Chrysene (**B**), and Phenanthrene (**C**) content (mg/kg of soil) in soils of the area studied.

**Figure 4 microorganisms-12-00080-f004:**
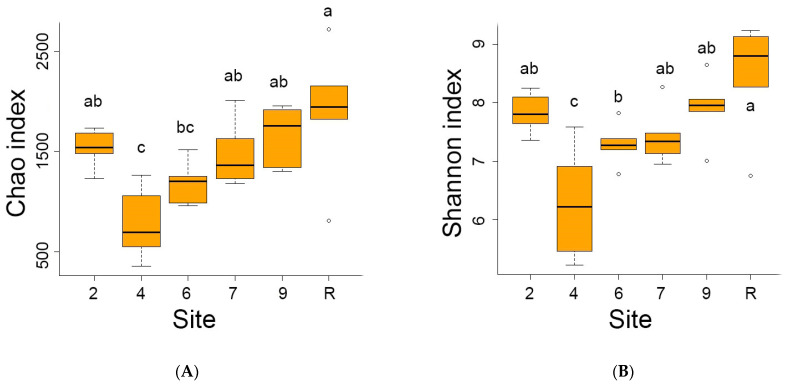
The Chao (**A**) and Schannon (**B**) diversity indices of bacterial communities on sites studied. Here and late significant differences between mean values (ANOVA, Student–Newman–Keuls test, *p* < 0.05) are indicated by different letters.

**Figure 5 microorganisms-12-00080-f005:**
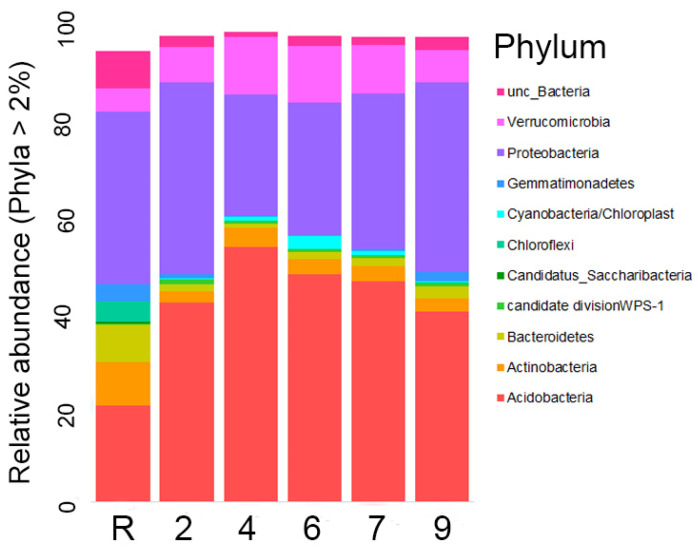
Relative abundance of the most numerous type of phyla representatives in the studied areas (the abundance of each phylum is more than 2%).

**Figure 6 microorganisms-12-00080-f006:**
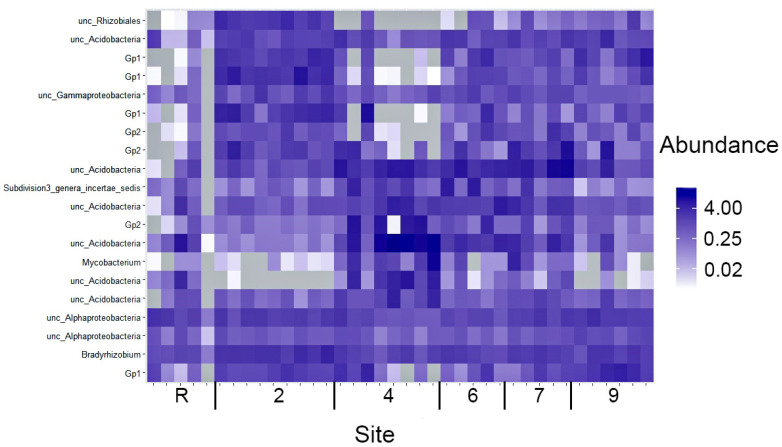
A heatmap of 20 OTU of the bacteria with maximal abundance (percentage) across samples of sites studied.

**Figure 7 microorganisms-12-00080-f007:**
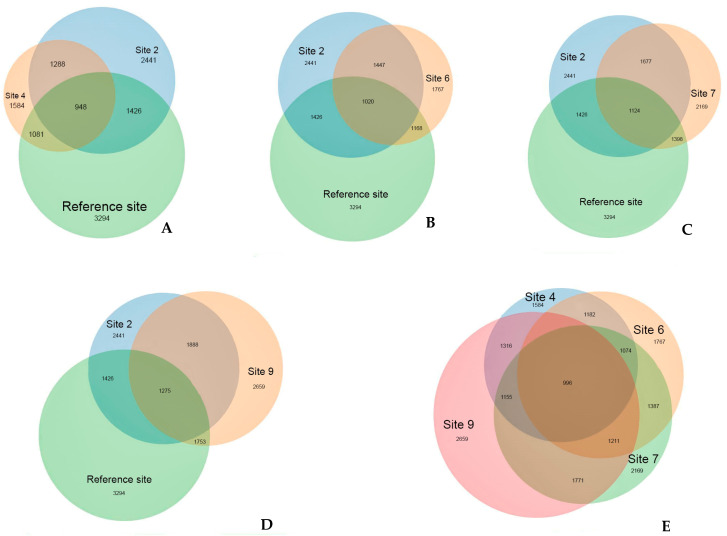
The number of common and unique OTUs detected at experimental sites. (**A**) reference site, sites 2 and 4; (**B**) reference site, sites 2 and 6; (**C**) reference site, sites 2 and 7; (**D**) reference site, sites 2 and 9; (**E**) sites 4, 6, 7, and 9. Figures are the number of OTUs.

**Figure 8 microorganisms-12-00080-f008:**
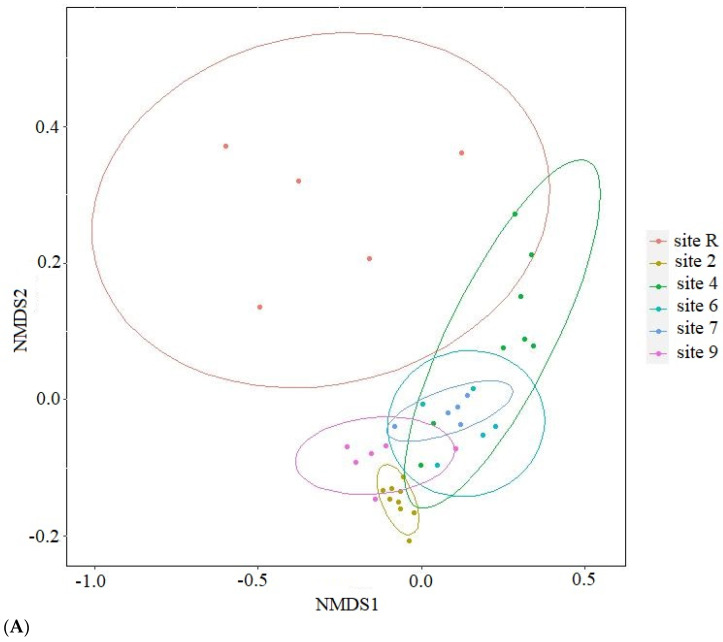
Results of ordination with (**A**) nonmetric multidimensional scaling based on Bray-Curtis distance; canonical correlation analysis for (**B**) all samples studied and (**C**) from oil-contaminated sites only. NO_3_—content of nitrogen in NO_3_ form; HM—sum of Cu, Zn, Pb and Cd in soil; oil—crude oil soil content; pH—soil pH.

**Figure 9 microorganisms-12-00080-f009:**
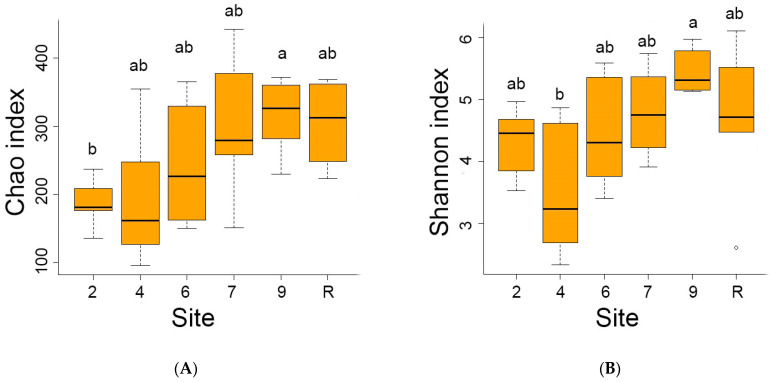
Chao (**A**) and Schannon (**B**) diversity indices of the fungi community on sites studied. Here and late significant differences between mean values (ANOVA, Student–Newman–Keuls test, *p* < 0.05) are indicated by different letters.

**Figure 10 microorganisms-12-00080-f010:**
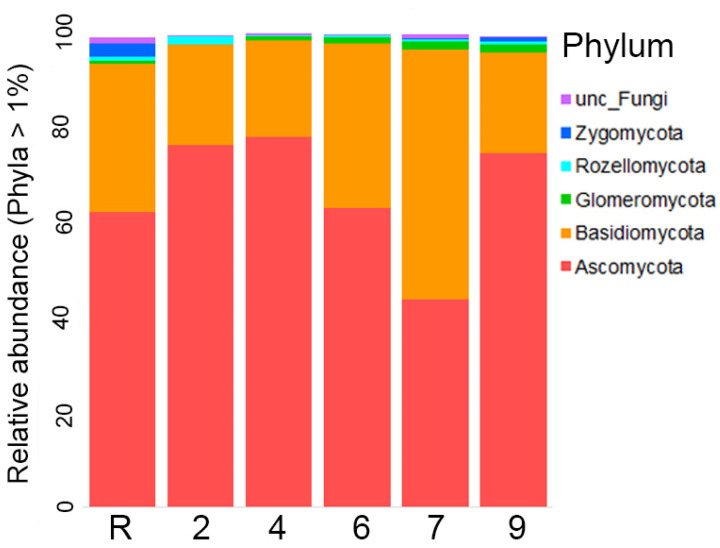
Relative abundance of different fungi phyla on sites studied (the abundance of each phylum is more than 1%).

**Figure 11 microorganisms-12-00080-f011:**
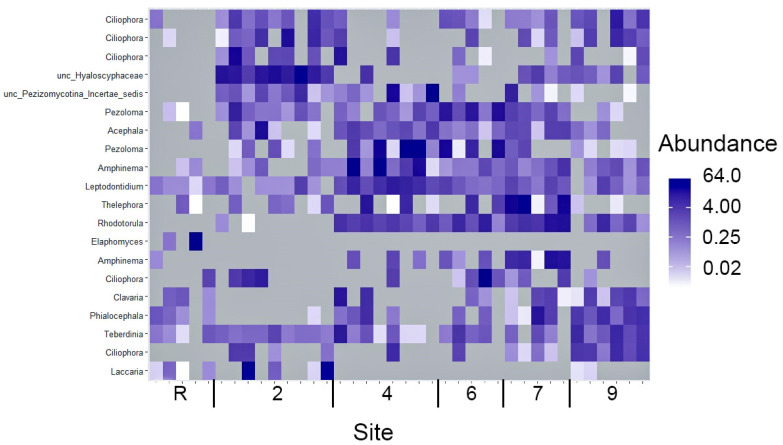
A heatmap of 20 OTU of the fungi with maximal abundance (percentage) across samples of sites studied.

**Figure 12 microorganisms-12-00080-f012:**
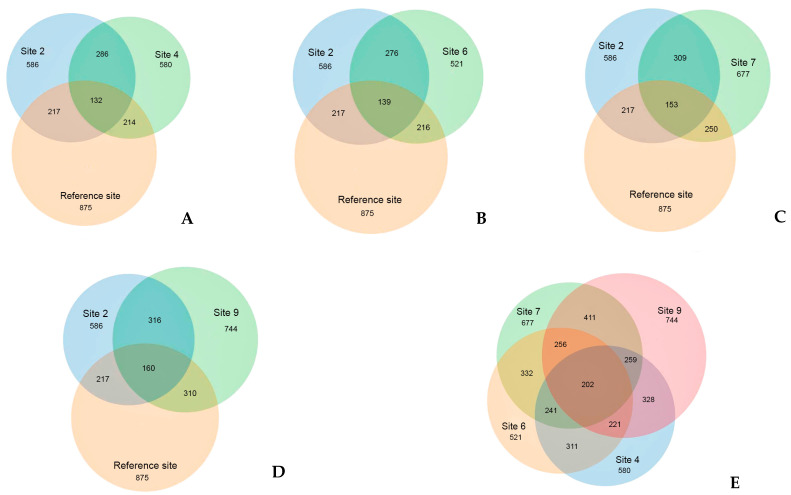
The number of common and unique OTU detected at experimental sites. (**A**) reference site, and sites 2 and 4; (**B**) reference site, and sites 2 and 6; (**C**) reference site, and sites 2 and 7; (**D**) reference site, and sites 2 and 9; and (**E**) sites 4, 6, 7, and 9. Figures are the number of OTUs.

**Figure 13 microorganisms-12-00080-f013:**
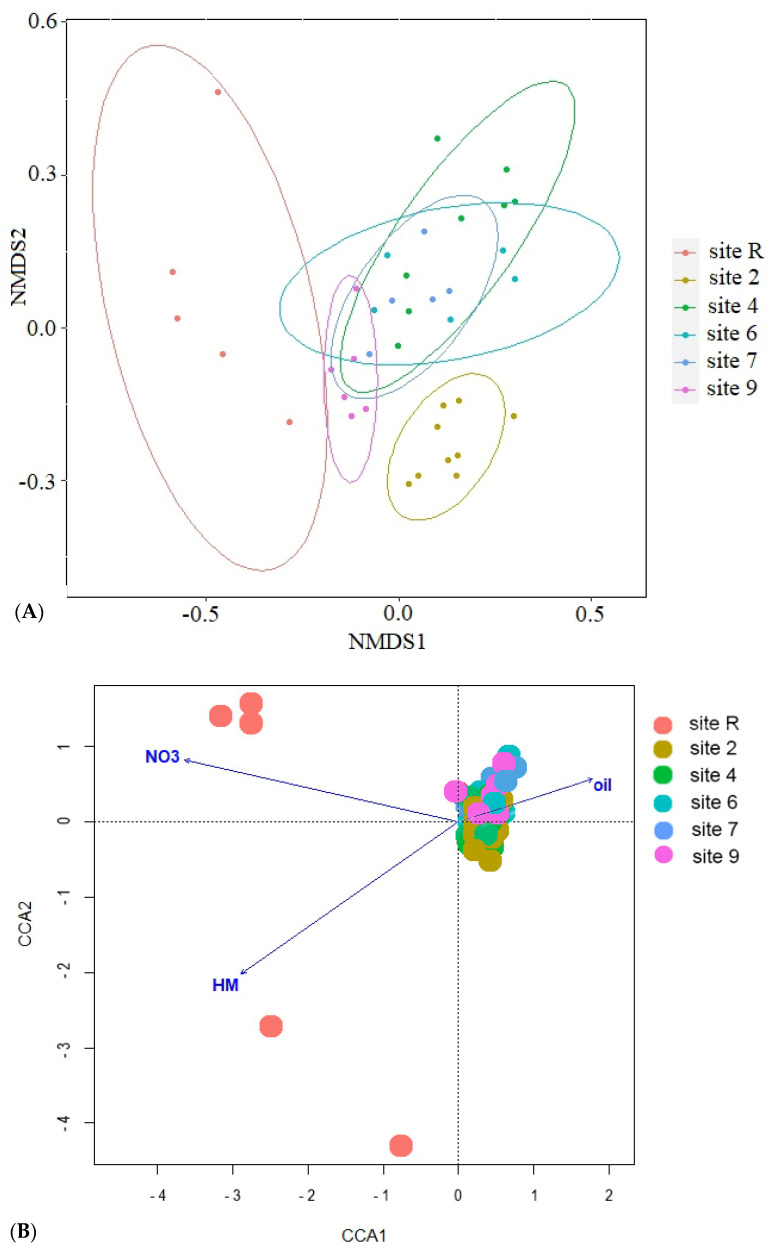
Results of ordination with (**A**) NMDS nonmetric multidimensional scaling based on Bray–Curtis distance; canonical correlation analysis for: (**B**) all samples studied and (**C**) from oil-contaminated sites only. NO_3_—content of nitrogen in NO_3_ form; HM—sum of Cu, Zn, Pb, and Cd in soil; oil—crude oil soil content; pH—soil pH; PAH—sum of polycyclic aromatic hydrocarbons concentrations in soil.

**Table 1 microorganisms-12-00080-t001:** Brief description of the remediation methods.

Site	Bio-Recultivation Method	Design	Fertilizers	Seeded Plants
Site 2	Control. Only mechanical oil removal. No biopreparations	-	No fertilizing	No seed plants
Site 4	Biopreparation “Universal” ^1^	yeasts *Rhodotorula glutinis* and bacteria *Rhodococcus egvi*, *Rhodococcus erythropolis*, *Pseudomonas fuorescens*	Mineral fertilizers ^3^	*Phleum pratense*, *Agrostis gigantea*, *Avena sativa*
Site 6	Biopreparation “Universal” ^1^, lignin sorbents, BAG	yeasts *Rhodotorula glutinis* and bacteria *Rhodococcus egvi*, *Rhodococcus erythropolis*, *Pseudomonas fuorescens*	Compost and mineral fertilizers	*Deschampsia cespitosa*
Site 7	Phytoremediation (without biopreparation)	-	Mineral fertilizers, dolomitic meal	*Avena sativa*, *Phleum pratense*
Site 9	Biopreparation “Roder” ^2^	bacteria *Rhodococcus ruber* and *Rhodococcus erythropolis*	Mineral fertilizers and lime	*Phalaris arundinacea*, *Phleum pratense*, *Avena sativa*
Site R	A reference site. Not subjected to oil contamination and other anthropogenic influences	-	-	-

^1^ “Universal”. Developer—The Institute of Biology Komi Science Center, Syktyvkar, Russia [23]. ^2^ “Roder”. Developer—Chemical Faculty of Moscow State University, Moscow, Russia [24]. ^3^ Mineral fertilizers—ANP (ammonium nitrate phosphate) fertilizer. Adapted from: [9].

**Table 2 microorganisms-12-00080-t002:** Soil pH, ammonia and nitrate nitrogen, and TPH (mg/kg of soil) in the study sites’ soil (mean ± SD).

Site	2	4	6	7	9	R ^1^
pH	5.1 ± 0.1 *	4.7 ± 0.2	4.7 ± 0.2	4.9 ± 0.1	5.2 ± 0.3 *	4.8 ± 0.3
N (NH_4_)	60.2 ± 18.5 *	82.9 ± 19.5 *	125.3 ± 12.9 *	71.1 ± 9.2 *	80.4 ± 22.3 *	808.6 ± 62.6
N (NO_3_)	8.1 ± 1.7 *	9.6 ± 1.0 *	14.0 ± 3.2 *	19.8 ± 4.4 *	20.4 ± 17.8 *	145.3 ± 103.4
Cu	8.0 ± 6.0	4.4 ± 1.6	4.6 ± 0.2	5.2 ± 0.8	5.8 ± 1.1	6.5 ± 1.8
Pb	3.7 ± 0.5 *	2.7 ± 0.3 *	4.2 ± 1.5 *	3.8 ± 1.2 *	4.1 ± 1.7 *	11.4 ± 2.0
Cd	0.3 ± 0.03	0.3 ± 0.08	0.3 ± 0.02	0.2 ± 0.05 *	0.4 ± 0.06	0.4 ± 0.09
Zn	7.5 ± 1.8 *	6.4 ± 2.9 *	5.0 ± 1.3 *	16.6 ± 4.3	10.6 ± 3.3 *	23.8 ± 6.8
TPH × 10^3^/Decrease ^2^	62.4 ± 29.2 */3.3	71.2 ± 93.8 */4.1	9.8 ± 3.1 */33.0	72.3 ± 28.6 */3.1	42.3 ± 46.4 */7.2	0.9 ± 0.7/3.3

*—difference with the reference site (R) values is significant at *p* < 0.05. ^1^—R—here and later reference site. ^2^—TPH ratio between mean values for the years 2002 (according to [22]) and 2019.

## Data Availability

The metataxonomics datasets generated during the current study are available in the NCBI repository (registration number PRJNA1052473). All other data generated during this study are included in the article.

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
