# Peer review of "Soil Microbiome in Conditions of Oil Pollution of Subarctic Ecosystems"

_microorganisms, 2023, doi:10.3390/microorganisms12010080_

Round 1

Reviewer 1 Report

Comments and Suggestions for Authors

Cold climate will limit the survival and metabolism of microorganisms. Therefore, the microbiological investigation of oil pollution remediation sites in cold areas is very necessary in this research. This manuscript is within the scope of the journal, but a number of ambiguities remain to be clarified.

1. The introduction needs to introduce the hazards of oil pollution, research progress in related fields, the importance of carrying out this research, and contribution to future work in a larger form.

2. Chapter 3.2 should not appear in "Results", it should appear in the form of an attachment in the additional materials as an explanation of the conditions in the study area.

3. In order to fully explain the differences in microbial species between remediation areas and unpolluted areas, it is necessary to analyze the correlation between bacterial and fungal diversity and soil chemical characteristics.

4. “Discussion” needs to be further strengthened on the basis of relevance scores

Comments on the Quality of English Language

Language should be carefully revised, if possible by a native English speaker.

Reviewer 2 Report

Comments and Suggestions for Authors

This is an interesting article. Unfortunately, it is spoiled by the poor English, which detracts substantially from ready understanding.

I should like to see more details given of the soil additions. The contents of each addition should be given (perhaps in a table); it is not enough to give a reference.

I am not sure all the photographs are required - most of them look the same to me. If there are differences obvious to botanists, these should be pointed out.

The results would benefit from a clear presentation of the differences between the results of the various treatments - perhaps a table?

Most importantly, the text must be read and corrected by an English speaker who understands the subject.

Comments on the Quality of English Language

I have already commented that the English requires extensive revision.

Round 2

Reviewer 1 Report

Comments and Suggestions for Authors

Agree

Comments on the Quality of English Language

Agree

Author Response

Dear reviewer, thank you very much for your work in reading our manuscript and making valuable comments. At this stage, we have carried out extensive editing of the English language and revised some sections of the manuscript.

Sincerely, Elena N. Melekhina

Reviewer 2 Report

Comments and Suggestions for Authors

It is still difficult to read this article because of the poor English. I believe that it could be an important piece of work, but it needs drastic editing and re-writing in good English. It should be much more succinct, emphasizing the important points - perhaps a series of bullet points could be used. I suggest that detailed discussion of microorganisms present is reduced to only those results that are strictly relevant. Do not give long lists of organisms detected if the significance of their presence cannot be explained, for example. 

Comments on the Quality of English Language

English is not good. It is not easy to read and this is a shame, as I believe that this could be a very interesting paper. Drastic shortening is needed, with the text giving only the most important conclusions and all other results being presented in tables and figures. This is the only way I can see the manuscript becoming acceptable, since this is a complex paper and it is obvious that the level of English is not sufficient for the detailed discussion that is being attempted. 

Author Response

Dear reviewer,

thank you very much for your work in reading our manuscript and making valuable comments.                   

At this stage, we have carried out extensive editing of the English language and revised some sections of the manuscript.

We tried to remove unnecessary detailed information, as a result of which the text of the manuscript was shortened. The annotation has been updated. The Introduction, as well as chapters 2.1, 2.2 and 2.3, have been edited and expanded. Chapter 3 and Chapter 4 have been revised and expanded.

Links have been made to additional sources that have been added to the References.

All corrections and additions have been highlighted in red font.

Sincerely,

Elena N. Melekhina
